# Increased Hepatocyte Growth Factor Secretion by Placenta-Derived Mesenchymal Stem Cells Improves Ovarian Function in an Ovariectomized Rat Model via Vascular Remodeling by Wnt Signaling Activation

**DOI:** 10.3390/cells12232708

**Published:** 2023-11-25

**Authors:** Hyeri Park, Dae Hyun Lee, Jun Hyeong You, Jin Seok, Ja-Yun Lim, Gi Jin Kim

**Affiliations:** 1Department of Bioinspired Science, CHA University, Seongnam-si 13488, Gyeonggi-do, Republic of Korea; hyeyeyeri@gmail.com (H.P.); ldh92426@gmail.com (D.H.L.); yjh950210@gmail.com (J.H.Y.);; 2PLABiologics Co., Ltd., Seongnam-si 13522, Gyeonggi-do, Republic of Korea; 3Department of Clinical Laboratory Science, Hyejeon College, Hongsung-gun 32244, Chungnam-do, Republic of Korea; jayun78@daum.net

**Keywords:** vascular remodeling, hepatocyte growth factor, Wnt signaling, follicular development, ovarian function

## Abstract

The vascular network contributes to the development of follicles. However, the therapeutic mechanism between vascular remodeling and ovarian functions is still unclear. Therefore, we demonstrated whether increased *HGF* by placenta-derived mesenchymal stem cells (PD-MSCs) improves ovarian function in an ovariectomized rat model via vascular remodeling by *Wnt* signaling activation. We established a half-ovariectomized rat model in which damaged ovaries were induced by ovariectomy of half of each ovary, and PD-MSCs (5 × 10^5^ cells) were transplanted by intravenous injection. Three weeks after transplantation, rats in all groups were sacrificed. We examined the secretion of *HGF* by PD-MSCs through culture medium. The vascular structure in injured ovarian tissues was restored to a greater extent in the PD-MSC transplantation (Tx) group than in the nontransplantation (NTx) group (* *p* < 0.05). The expression of genes related to *Wnt* signaling (e.g., *LRP6*, *GSK3β*, *β-catenin*) was significantly increased in the Tx group compared to the NTx group (* *p* < 0.05). However, the expression of genes related to vascular permeability (e.g., *Asef*, *ERG3*) was significantly decreased in the Tx group compared to the NTx group (* *p* < 0.05). Follicular development was improved in the Tx group compared to the NTx group (* *p* < 0.05). Furthermore, to evaluate vascular function, we cocultivated PD-MSCs after human umbilical vein endothelial cells (HUVECs) with lipopolysaccharide (LPS), and we analyzed the vascular formation assay and dextran assay in HUVECs. Cocultivation of PD-MSCs with injured HUVECs enhanced vascular formation and decreased endothelial cell permeability (* *p* < 0.05). Also, cocultivation of PD-MSCs with explanted ovarian tissues improved follicular maturation compared to cocultivation of the Wnt inhibitor-treated PD-MSCs with explanted ovarian tissues. Therefore, *HGF* secreted by PD-MSCs improved ovarian function in rats with ovarian dysfunction by decreasing vascular permeability via Wnt signaling.

## 1. Introduction

An elaborate vascular network developed in the ovary delivers oxygen, nutrients, cytokines and hormone substrates to the follicle, playing a key role in follicular maturation as well as ovarian function [1,2]. Initiation of follicular development is activated by the receipt of cytokines through blood vessels in contact with the follicle. During follicle maturation, arteries and veins sprout into the internal theca cells layer, but do not penetrate the basement membrane and granulosa cells. Granulosa cells and oocytes obtain nutrients, hormones and cytokines by spreading across the theca cell layer [3,4]. Many proangiogenic factors are known to regulate follicular development. In particular, vascular endothelial growth factor (*VEGF*) regulates thecal angiogenesis and supports follicular development [5]; additionally, platelet-derived growth factor (*PDGF*), which is crucial for blood vessel stability, stimulates the proliferation of theca cells in antral follicles [6]. The balanced expression of these factors is critical for follicular development during follicular initiation, growth, maturation and selection due to their role in regulating the follicular vasculature [7,8]. Because the ovarian vasculature and its function can be damaged and modulated by the inflammatory response, damage to the ovarian vasculature results in ovarian dysfunction that leads to follicular nutrient, hormone and cytokine deficiencies [9]. In the ovary, *CD68*-positive macrophages are localized near the vascular and theca cell layers. Ovarian immune cells, such as macrophages, maintain homeostasis and regulate ovarian function through the vasculature [10]. Orostica et al. and Saccon et al. reported that macrophage infiltration occurs around the follicles in conditions characterized by ovarian abnormalities, such as polycystic ovarian syndrome (PCOS), and as the ovaries age [11,12]. Therefore, proinflammatory factors are upregulated, and an abnormal intraovarian macrophage phenotype is detected in the ovary during aging. Chronic inflammation is known to contribute to not only age-associated endothelial dysfunction in the ovary but also female reproductive diseases of abnormal self-recognition by the immune system, such as premature ovarian failure (POF) [13,14]. Recently, Kalantaridou et al. reported that young women with POF have symptoms of endothelial dysfunction that are not observed in women of similar age with normal ovarian function [15]. For the treatment of ovarian dysfunction related to aging and inflammation, hormone replacement therapy (HRT) is commonly used. However, since HRT has short-term effects and is associated with an increased risk of various conditions (e.g., osteoporosis, breast cancer and cardiovascular) [16,17], phosphatase and tensin homolog (*PTEN*) inhibitors and therapies based on stem cells have been studied to overcome the limitations of HRT. While *PTEN* inhibitors activate the growth of primordial follicles [18], stem cell therapy exerts a therapeutic mode of action (MoA) on ovarian function, improving overall ovarian function [19].

Cytokines secreted by stem cells have been reported to improve ovarian function in several animal models with ovarian dysfunction through therapeutic mechanisms, such as the regulation of sex hormones, a reduction in inflammation, a reduction in oxidative stress and the prevention of apoptosis [19]. *HGF*, which is a vascular modulator, was reported to localize in theca cells in the ovary and regulate follicular development. *HGF* is known to regulate hormone levels as well as proliferation of granulosa cells in ovarian. Mi et al. reported that *HGF* secreted from human umbilical cord mesenchymal stem cells (hUC-MSCs) increased the activity of primordial follicles by the *PI3K*-*AKT* pathway in a mouse model with POF. Ding et al. demonstrated that human adipose-derived stem cells enhance ovarian function through the inhibition of apoptosis by secreting *HGF* [20,21,22]. Generally, the activation of the *Wnt* signaling pathway is acknowledged to occur through *HGF* [23,24], and the *Wnt*/*β-catenin* pathway is recognized for its regulation of multiple aspects of vascular remodeling, inflammation and metabolism [25]. Accordingly, vascular degeneration was reported in endothelial cells in which *Wnt* was knocked down, and inactivation of the *Wnt*/*β-catenin* pathway increased immune cell infiltration in endothelial tissues [26,27]. In addition, *Wnt* signaling is critically involved in ovarian function associated with follicular development, formation of corpus luteum, steroidogenesis and fertility [28], and it has been reported that *Wnt5A* expression is decreased in the theca cells of PCOS patients [29]. Additionally, Wang et al. demonstrated in *Wnt2*-overexpressing granulosa cells that *Wnt2*/*β-catenin* signaling regulates the proliferation of granulosa cells through the expression of *β-catenin* and *GSK3β*, and *Wnt2* was identified in granulosa cells throughout all stages of follicular development in immature rat ovaries [30]. *Wnt* signaling has also been suggested to have a negative effect on follicular development due to its effects on the hormonal environment [31].

In our previous report, we demonstrated that transplanted PD-MSCs reduce oxidative stress and improve ovarian function through vascular remodeling via *VEGF* signaling and pericyte recruitment via *PDGF* signaling [32,33]. Nevertheless, the mechanisms underlying the effects of *Wnt* signaling on vascular permeability in the ovary remain uncertain. To confirm the activation of *Wnt* signaling in ovarian tissues with transplanted PD-MSCs, we analyzed the expression of genes related to *Wnt* signaling downstream and expression of genes (e.g., *Asef*, *ERG*) associated with vascular permeability among *HGF* and *Wnt* signaling. To address the debate about the various effects of *Wnt* signaling on ovarian function, we prepared an explanted ovarian system that was treated with BIO, which is a *Wnt* inhibitor. Hence, the aim of this study was to determine whether the expression of *HGF* by PD-MSCs improves ovarian function in an ovariectomized rat model via vascular remodeling by *Wnt* signaling activation.

## 2. Materials and Methods

### 2.1. Cell Culture

Approval for obtaining human placental samples for research purposes was granted by the Institutional Review Board of CHA Gangnam Medical Center, Seoul, Republic of Korea (IRB-07-18). Shortly, as outlined in a prior report, PD-MSCs were extracted from the chorionic plate of the placenta [34]. PD-MSCs were cultivated in alpha-minimum essential medium (α-MEM; HyClone, Logan, UT, USA) involving 10% fetal bovine serum (FBS; Gibco-BRL, Oklahoma, USA), 1% penicillin/streptomycin (P/S; Gibco-BRL, Oklahoma, USA), 25 ng/mL *FGF-4* (PeproTech, Cranbury, NJ, USA), and 1 µg/mL heparin (Sigma-Aldrich, Saint Louis, MO, USA) at 37 °C incubator under 5% CO_2_. Harvested PD-MSCs were labeled with a PKH67 Fluorescent Cell Linker Kit (Sigma-Aldrich, Saint Louis, MO, USA). Then, 5 × 10^5^ cells of PD-MSCs were injected intravenously in the transplanted group.

### 2.2. Construction of the Half-Ovariectomized Rat Model

Seven-week-old female Sprague-Dawley rats (Orient Bio, Inc., Seongnam-si, Republic of Korea) were kept in an air-controlled animal facility. The rats were housed in pairs within plastic cages containing corn cob bedding and were provided unrestricted access to standard commercial food and tap water. There was a 12 h light-dark cycle, and the temperature was set at 21 °C. The Institutional Animal Care and Use Committee of CHA University, Seongnam-si, Republic of Korea (IACUC-190048) gave its approval to the animal models and experimental protocols. Before the operation, each individual rat was induced into general anesthesia via an abdominal injection of avertin (2,2,2-tribromoethanol, Sigma-Aldrich, Saint Louis, MO, USA). A half ovariectomy was conducted by surgically excising half of each ovary, and hemorrhage of the surgical sites was arrested with tissue adhesive (3M Corporate, Maplewood, MN, USA). Each group, which included the normal, NTx and Tx groups, consisted of 6–7 rats. One week after operating, 5 × 10^5^ cells of PD-MSCs (10–13 passages) labeled with a PKH67 Cell Linker Kit (Sigma-Aldrich, Saint Louis, MO, USA) were intravenously injected via the tail vein. Rats from all groups were sacrificed 3 weeks after transplantation, and ovarian tissues and blood samples were collected from them.

### 2.3. RNA Isolation and Quantitative Real-Time Polymerase Chain Reaction

Total RNA was extracted from total RNA from the rat ovarian tissues using TRIzol reagent (Ambion, Austin, TX, USA; Thermo Fisher Scientific, Waltham, MA, USA) according to the manufacturer’s instructions. The concentration of total RNA was determined using a NanoDrop spectrophotometer (Thermo Fisher Scientific, Waltham, MA, USA). Reverse transcription of total RNA into cDNA was accomplished using Superscript III reverse transcriptase (Invitrogen, Waltham, MA, USA). The PCR setting used for cDNA synthesis were as follows: 5 min at 65 °C, 1 min at 4 °C, 1 h at 50 °C, and 15 min at 72 °C. The cDNA was subsequently amplified by qRT-PCR reaction using SYBR Ex Taq (Roche, Basel, Switzerland) under the following conditions: 5 s at 95 °C and 40 cycles of 95 °C for 5 s and 60 °C for 30 s. Appendix A contain a list of the qRT-PCR primer sequences that were employed. *GAPDH* served as an internal control for normalization, and each sample was assessed in triplicate.

### 2.4. Ovarian Explant Ex Vivo Culture

Indirect cocultivation system was established employing an insert system (Falcon, New York, NY, USA) to confirm the paracrine effect of PD-MSCs in ovarian tissues. To cultivate ovarian tissue, Matrigel (Corning, New York, NY, USA) was coated in a 24-well culture plate and incubated for 3 h. Then, the ovaries of 7-week-old female Sprague–Dawley rats were dissected, washed with saline, and DPBS containing 1% penicillin/streptomycin. Disectioned ovary was explanted on a Matrigel-coated 24-well culture plate. 1 × 10^4^ cells of PD-MSCs were cocultured with the disectioned ovarian tissue on an 8 µm pore size insert (Falcon, New York, NY, USA). Then, BIO (1 µM; Stemgent, San Diego, CA, USA), which is *GSK3β* inhibitor, or IWP2 (1 µM; Stemgent), which is *LRP6* inhibitor, were added to the medium. After 24 h, the disectioned ovarian tissues and supernatant were collected, and these collected samples were analyzed.

### 2.5. Western Blotting

Rat ovarian tissues from each group were homogenized with liquid nitrogen. The powdered rat ovarian tissues were lysed with RIPA buffer (Sigma-Aldrich, Saint Louis, MO, USA) containing a protease inhibitor cocktail (Roche) and phosphatase inhibitor cocktail (genDEPOT, Katy, TX, USA). Equal concentrations of extracted protein were subjected to 8–12% sodium dodecyl sulfate-polyacrylamide gel electrophoresis (SDS-PAGE). The separated proteins were transferred onto polyvinylidene difluoride (PVDF) membranes (Bio-Rad Laboratories, Hercules, CA, USA) using a Transfer Turbo system (Bio-Rad Laboratories). After transfer, the membranes were washed with 1X Tris-buffered saline-Tween 20 (TBS-T) for 5 min. The membranes were blocked in blocking buffer (5% BSA based on 1X TBS-T) at room temperature for 1 h. After blocking, the membranes were incubated with primary antibody in 2% BSA at the dilutions listed in Appendix A for an overnight incubation at 4 °C. After incubation, the membranes were washed with 1X TBS-T, and incubated with secondary antibody in 2% BSA at the dilutions for 1 h at room temperature. After 1 h, the membranes had been washed with 1X TBS-T. Then, they were treated with the reagent from a Clarity Western ECL kit (Bio-Rad Laboratories, Hercules, CA, USA) for 3 min at room temperature. The ChemiDoc XRS+ imaging system (Bio-Rad Laboratories, Hercules, CA, USA) was used to detect the protein bands. The ImageJ program (https://imagej.net/ij/, Wayne Rasband, Bethesda, MD, USA) was used to analyze the protein bands. The comparative measure of gene expression utilized was the fold change in intensity.

### 2.6. Nuclear Fractionation

The 20 mg of ovarian tissues from each group were diluted with CER I solution (100 µL, NE-PERTM Nuclear and Cytoplasmic Extraction Reagents, Thermo Fisher Scientific, Waltham, MA, USA) with 1X phosphatase inhibitor, and incubated on ice for 10 min. Then, CER II solution (5 µL, NE-PERTM Nuclear and Cytoplasmic Extraction Reagents, Thermo Fisher Scientific, Waltham, MA, USA) was introduced and thoroughly mixed using vortexing. Following the vortexing process, the samples were chilled on ice for 1 min and centrifuged (~16,000× *g*) for 5 min. After centrifugation, the supernatants (cytoplasmic) were transferred to fresh prechilled tubes. The pellets were suspended in NER solution (50 µL, NE-PERTM Nuclear and Cytoplasmic Extraction Reagents, Thermo Fisher Scientific, Waltham, MA, USA) with 1X phosphatase inhibitor. The samples were vortexed briefly for 10 s every 10 min over a total duration of 50 min on ice. Then, the samples were centrifuged at maximum speed (~16,000× *g*) in a microcentrifuge for 10 min. The supernatant, which held the nuclear extract, was transferred to clean prechilled tubes.

### 2.7. ELISA

The blood samples were obtained by harvesting the aortas from each rat from every group (normal, NTx and Tx groups). Using a blood collection tube (Vacutainer; BD Biosciences, San Jose, CA, USA), individual serum samples were separated from whole blood. All serum was stored at −80 °C deepfreezer. The interleukin-6 (*IL-6*), interleukin-10 (*IL-10*) (Abcam, Waltham, MA, USA) and AMH (Elabscience Biotechnology, Waltham, MA, USA) activity in the serum were analyzed using ELISA kits in accordance with the manufacturer’s instructions. Briefly, an equivalent volume of the sample was introduced to plates coated with specific antibodies. Then, horseradish peroxidase (HRP)-conjugates were added and incubated at 37 °C. A microplate reader (BioTek, Winooski, VT, USA) was utilized to measure antibody activity after the substrates had been introduced and left to develop in the dark.

### 2.8. TUNEL Assay

For the detection of apoptosis through DNA fragmentation, the ovarian tissues were stained with reagent from a TUNEL assay kit (Abcam, Waltham, MA, USA) according to guidelines provided by the manufacturer. The 4 µm ovary tissue sections were deparaffinized in a 60 °C dry oven for 1 h. Then, the sections were deparaffinized using xylene and ethanol. The deparaffinized tissues were washed using 1X TBS-T. The ovarian tissues were permeabilized by proteinase K and treated with 3% H_2_O_2_ to block endogenous peroxidases. Then, the ovarian tissues were labeled with TdT enzyme inTdT equilibration buffer. The specimens were stained with DAB solution for development. Finally, for counterstaining, the ovarian tissues were stained with a methyl green counterstain solution and mounted on a glass coverslip using organic mounting medium. To analyze the positive control, the remaining ovarian tissue was treated with 1 µg/µL DNase I in TBS/1 mM MgSO_4_ at room temperature for 20 min. After all staining procedures, the tissues were analyzed with the 3D HISTECT program (The Digital Pathology Company, Budapest, Hungary).

### 2.9. Immunohistochemistry Staining

The 4 µm ovary tissue sections were deparaffinized in a 60 °C dry oven for 1 h. After cooling for 1 h, the tissue sections were deparaffinized using xylene and ethanol. Deparaffinized tissues underwent antigen retrieval through the EDTA (eLbio, Seongnam-si, Republic of Korea) reaction, followed by gradual cooling with tap water. The ovarian tissues were treated with a peroxide blocking solution containing 3% H_2_O_2_ in methanol for 10 min after being cleaned with distilled water (D.W.). The ovarian tissues were cleaned with D.W. and treated with primary antibodies in diluent buffer (Dako, Carpinteria, CA, USA) overnight at 4 °C. Negative control of ovarian tissue was treated with diluent buffer without primary antibodies. Following the elimination of the unbound primary antibody, Real EnVision HRP rabbit/mouse secondary antibody (Dako, Carpinteria, CA, USA) was used to incubate the tissues on the slides for 1 h at room temperature. The slides were treated with DAB and counterstained with Mayer hematoxylin (Dako, Carpinteria, CA, USA). After staining, the slides were rinsed with tap water. The slides were dehydrated using ethanol and xylene. The stained tissues were scanned and analyzed using the 3D HISTECT program (The Digital Pathology Company, Budapest, Hungary).

### 2.10. Immunofluorescence Staining

The 7 µm of ovarian tissue section from frozen blocks were fixed using methanol for 20 min. Following air drying, the ovarian tissues were rinsed with precooled 1× phosphate-buffered saline (PBS) at room temperature 3 times for 5 min each. After removing the 1× PBS at the tissue edge, the tissues were exposed to a blocking solution (Dako, Carpinteria, CA, USA) at room temperature for 1 h and incubated with primary antibodies in diluent buffer at 4 °C overnight. Negative control of ovarian tissue was treated with diluent buffer without primary antibodies. After incubation, the tissues were washed with 1× PBS at room temperature 3 times for 5 min each and were subsequently incubated with secondary antibody in diluent buffer at room temperature for 1 h. The ovarian tissues were washed with 1X PBS at room temperature for 5 min 3 times each. Then, the tissues were counter-stained and mounted with mounting medium containing DAPI (Vectashield, Burlingame, CA, USA). The fluorescence microscopy (Zeiss LSM 780, Oberkochen, Germany) was used to observe the tissues at 40× and 63× magnification. Every section of each slide was examined, and representative images were captured.

### 2.11. H&E Staining for Follicle Counting

The fixed ovarian tissues using 10% neutral buffered formalin (BBC, Washington, DC, USA) were embedded in paraffin. The paraffin blocks were sequentially sectioned into 4 µm sections of ovarian tissue. The ovarian tissue sections were deparaffinized in a 60 °C dry oven for 1 h and cooled in room temperature for 1 h. Then, the tissues were deparaffinized with xylene and ethanol. Deparaffinized tissues were rinsed with tap water. The slides were counter-stained with Harris hematoxylin (Leica Biosystems, Wetzlar, Germany) for 7 min, dipped in 0.1% HCl for 2 s and stained with alcoholic eosin Y solution (Sigma-Aldrich, Saint Louis, MO, USA). The stained slides were scanned to encompass the entire ovarian tissues using 3D HISTECH (The Digital Pathology Company, Budapest, Hungary). Follicles were counted by selecting a slide every 100 µm of the slides that were serially sectioned. According to previous reports, we classified and counted follicles as primordial, primary, secondary, antral and atresia follicles [35]. The follicle counts were analyzed, and three or more persons performed follicle counting for verification.

### 2.12. HUVECs Vascular Formation Assay to Assess Vascular Function

HUVECs were cultured using ECM (Science Cell, Los Angeles, CA, USA) medium at 37 °C in a 5% CO_2_ incubator. The 3 × 10^4^ cells of HUVECs were seeded on a Matrigel-coated 24-well culture plate. After 24 h, the cells underwent treatment with 20 ng/mL lipopolysaccharide (LPS) for 24 h. After 24 h, the medium was changed to the ECM medium without LPS. 1 × 10^4^ cells of PD-MSCs were seeded on 8 µm pore size inserts (Falcon, New York, NY, USA). After cocultivation for 24 h, the inserts were removed. Next, the medium was changed with medium including 2 µg/mL Alexa Fluor 488-conjugated Ac-LDL (Invitrogen, Waltham, MA, USA). The HUVECs were stained by medium including Alexa Fluor 488-conjugated Ac-LDL for 24 h at 37 °C in a 5% CO_2_ incubation environment. After LDL staining, the HUVECs were washed with DPBS. The vessel branching length was quantified using the ImageJ program.

### 2.13. Dextran HUVECs Permeability Assay to Assess Vascular Function

HUVECs were incubated at 37 °C in a 5% CO_2_ environment with ECM medium (Science Cell, Los Angeles, CA, USA). Matrigel was coated on an insert with a 0.4 µm pore size (Falcon, New York, NY, USA) for 3 h. After coating, 3 × 10^4^ cells of HUVECs were seeded on the insert. After 24 h, the HUVECs were treated medium with 20 ng/mL LPS. After treatment for 24 h, the medium was substituted with LPS-free medium. PD-MSCs were planted in 24-well plates (Falcon New York, NY, USA). After cocultivation for 24 h, the inserts were replaced to clean wells containing basal medium. Then, dextran (Sigma-Aldrich, Saint Louis, MO, USA) was treated to the inserts. After 20 min, the medium from a 24-well plate was transferred to a black 96-well assay plate (Costar, Washington, DC, USA). The plate was measured using Tecan assay at the absorbance at 485 nm (excitation) and 535 nm (emission).

### 2.14. Statistical Analysis

Every experiment was carried out in triplicate or more. The results are expressed as the mean ± standard error. Statistical analysis was conducted using GraphPad Prism 5.0 (GraphPad Software, Inc., San Diego, CA, USA), employing one-way ANOVA followed by Tukey’s multiple comparisons test. Statistical significance was attributed to differences with *p* value less than 0.05.

## 3. Results

### 3.1. PD-MSCs Transplantation Prevented Inflammation in the Ovarian Tissues of Half Ovariectomized Rats

To confirm that inflammation occurred in the acutely injured ovary, we analyzed macrophage infiltration in the antral follicles of the ovaries. The gene expression of *CD68*, which is known to be involved in macrophage infiltration, caused the production of atresia follicles in normal ovarian tissues. The positive *CD68* signal was localized near capillaries and theca cells in the ovarian tissues upon acute injury. Macrophage infiltration was markedly increased in the nontransplantation (NTx) group compared to the transplantation (Tx) group (Figure 1A,B; * *p* < 0.05). As PD-MSCs engrafted into the ovaries, macrophage infiltration near the capillary nearby theca cells of mature follicles considerably decreased. The mRNA expression of *hAlu* was absent in the NTx group but was detected in the Tx group (Figure 1C; * *p* < 0.05). Additionally, we analyzed the concentrations related to inflammation in the serum and homogenized ovarian tissues of the rats. The secretion of *IL-6*, which is a proinflammatory cytokine, exhibited a significant increase in the NTx group compared to the normal group and a decrease in the Tx group compared to the NTx group (Figure 1D; * *p* < 0.05). The secretion of *IL-10*, which is an anti-inflammatory cytokine, was significantly decreased in the NTx group compared to the normal group and was not significantly different between the NTx and Tx groups (Figure 1E; * *p* < 0.05). The protein expression of *IL-6* was markedly increased in the NTx group compared to the normal group and was significantly decreased in the Tx group compared to the NTx group (Figure 1F; * *p* < 0.05). Furthermore, the protein expression of *TNFα* was considerably increased in the NTx group compared to the normal group and significantly decreased in the Tx group compared to the NTx group (Figure 1G; * *p* < 0.05). These data suggest that transplanted PD-MSCs reduced macrophage infiltration into the ovary upon acute injury.

To confirm that inflammation induces apoptosis in the ovary, we conducted a TUNEL assay to identify nuclear DNA fragments in ovarian tissues from rats following acute ovarian injury (Figure 1H). The proportion of detected apoptotic cells was notably increased in the NTx group compared to the normal group. However, the apoptotic signal was significantly decreased in the Tx group compared to the NTx group (Figure 1I; * *p* < 0.05). In addition, we confirmed cytotoxicity in the serum from each individual rat through a lactate dehydrogenase (LDH) assay. There was an elevated level of cytotoxicity in the NTx group compared to the normal group. Its level was significantly decreased in the Tx group compared to the NTx group (Figure 1J; * *p* < 0.05). These data suggest that inflammation-induced apoptosis was decreased through the transplantation of PD-MSCs.

### 3.2. Activated Wnt Signaling Improved Vascular Remodeling via Increased HGF Secretion by PD-MSCs in the Ovarian Tissues of Half Ovariectomized Rats

To assess the structure of blood vessels following PD-MSCs transplantation, we quantified the vessels and measured both the artery thickness and the lumen area of the vessels (Figure 2A). While there was no significant difference in the number of blood vessels between the groups, the thickness of the artery exhibited a significant increase in the NTx group compared to the normal group. In the Tx group, the vascular structure in the ovary was similar to that in the normal group. (Figure 2B; * *p* < 0.05). Also, we analyzed the mRNA expression of factors associated with vascular structure. The Hippo pathway, which is induced by *Wnt* signaling, is known to regulate homeostasis, endothelial cell shape and immunity suppression [36]. The expression of the transcriptional regulator yes1-associated (*YAP*) was notably decreased in the NTx group compared to the normal group and significantly elevated in the Tx group compared to the NTx group (Figure 2C; * *p* < 0.05). These data indicated that transplanted PD-MSCs changed the structure of blood vessels in the injured ovary.

We hypothesized that *Wnt* signaling is activated by *HGF* secreted via the paracrine activity of PD-MSCs, thereby enhancing ovarian function by restructuring the vasculature of the ovary [24] and analyzed measurement of cytokine levels using Profiler Human XL cytokine arrays (Figure 2D). Also, we observed a significant increase in the *HGF* level in the supernatant of PD-MSCs compared to basal medium and WI-38 (Figure 2E). The protein expression of *c-Met*, which is an *HGF* receptor, was significantly reduced in the NTx group compared with the normal group but was not significantly different between the NTx and Tx groups (Figure 2F; * *p* < 0.05).

*Wnt* signaling is known to affect vascular remodeling and exert an anti-inflammatory effect [37]. To demonstrate *Wnt* signaling activation, we analyzed the expression of low-density lipoprotein receptor-related protein 6 (*LRP6*) and glycogen synthase kinase-3 beta (*GSK3β*) were notably decreased in the NTx group compared with the normal group and increased in the Tx group compared with the NTx group (Figure 2G,H; * *p* < 0.05). To demonstrate *β-catenin* expression as a marker of *Wnt* signaling downstream in vessels of the ovary, immunofluorescence staining was performed. The expression of *β-catenin* was trans-localized in the vessels of the ovarian tissues (Figure 2I). The quantitative ratio of *β-catenin*-positive signal versus *CD31*-positive signal was substantially decreased in the NTx group compared to the normal group and increased in the Tx group compared to the NTx group (Figure 2J; * *p* < 0.05). These data indicated that transplanted PD-MSCs activated *Wnt* signaling in the vessels of the injured ovaries of rats.

Rho guanine nucleotide exchange factor (*Asef*), which is a gene related to *HGF* and *Wnt* signaling, is known to regulate endothelial cell permeability [38]. To confirm the improvement in vascular function induced by activated *Wnt* signaling, the expression of *Asef* in blood vessels in the ovary was analyzed (Figure 2K). The quantitative ratio of *Asef*-positive signal versus *CD31*-positive signal was significantly decreased in the NTx group compared to the normal group and notably increased in the Tx group compared to the NTx group (Figure 2L; * *p* < 0.05). These data indicated that *Wnt* signaling improves vascular function (e.g., vascular permeability) through increased *HGF* expression by transplanted PD-MSCs.

### 3.3. Effect of PD-MSCs on Follicular Development in the Ovarian Tissues of Half Ovariectomized Rats

To demonstrate that ovarian function was ameliorated by vascular remodeling, we analyzed the anti-Mullerian hormone (AMH) levels in the serum of individual rats with injury. The level of AMH, which is involved in ovarian reserve, was significantly decreased in the NTx group compared to the normal group and significantly increased in the Tx group compared to the NTx group (Figure 3A; * *p* < 0.05). Furthermore, we examined the mRNA expression of genes associated with follicular development in the ovarian tissues of the injured rats. The mRNA expression of epidermal growth factor receptor (*EGFR*), which regulates oocyte maturation during follicular development, was significantly decreased in the NTx group compared to the normal group; however, there was no significant difference between the NTx group and Tx group (Figure 3B; * *p* < 0.05). The mRNA expression of bone morphogenetic protein 15 (*BMP15*), which is associated with the maturation of follicular development, was markedly decreased in the NTx group compared to the normal group, but their expression showed a significant increase in the Tx group compared to the NTx group (Figure 3C; * *p* < 0.05). Especially, *BMP15* was localized in mature follicles (such as secondary and antral follicles) in the ovarian tissues (Figure 3D) and the expression of these genes in granulosa cells within the follicles considerably decreased in the NTx group compared to the normal group and significantly increased in the Tx group compared to the NTx group (Figure 3E; * *p* < 0.05).

To confirm the effect of PD-MSCs on follicle development, we stained the ovarian tissues and quantified the follicles in the stained samples using the 3D HISTECH program based on the stage of follicular development (Figure 3F). PD-MSCs transplantation protected the follicles at each stage (e.g., primordial, primary, secondary and antral follicles). The maturation stage of follicular development (e.g., secondary, antral follicles) did not proceed in the NTx group but did in the normal group, but maturation development occurred in the NTx group when PD-MSCs were transplanted. Additionally, there was a significant increase in atresia follicles in the NTx group compared to the normal group and a significant decrease in the Tx group compared to the NTx group (Figure 3G, Table 1; * *p* < 0.05). These results indicated that transplantation of PD-MSCs improved follicular development, including maturation of granulosa cells in injured ovarian tissues.

To confirm the factors related to the maturation stage of follicular development, after treating KGN cells with LPS to induce inflammation, we cocultivated the KGN cells with PD-MSCs (Figure 3H). The mRNA expression of *EGFR* was remarkably decreased in the non-cocultivated group compared to the normal group. These mRNA expression was increased in the PD-MSC cocultivation group compared to the non-cocultivated group (Figure 3I; * *p* < 0.05). The mRNA expression of *BMP15* was importantly decreased in the non-cocultivated group compared to the normal group and considerably increased in the PD-MSC cocultivation group compared to the non-cocultivated group (Figure 3J; * *p* < 0.05). These results suggest that PD-MSCs improved follicular development including maturation of granulosa cells in injured ovarian tissues.

### 3.4. Effect of PD-MSCs on Follicular Development via Wnt Inhibitor Treatment of Ex Vivo Ovarian Tissues

To demonstrate the effect of PD-MSCs on the Wnt signaling pathway in injured ovarian tissues, we performed an ex vivo experiment using treatment with BIO and IWP2, which are the *GSK3β* inhibitor and *LRP6* inhibitor, because *GSK3β* and *LRP6* are known to be key factors regulating the *Wnt*/*β-catenin* pathway. Since *GSK3β* is multiply regulated in *Wnt* signaling, validation was performed using IWP2, a clear *Wnt* inhibitor [39]. First, we confirmed the mRNA expression of anti-inflammation (e.g., *IL-10*) and pro-inflammation (e.g., *IL-6*) in ovarian tissues. The mRNA expression of *IL-10* was considerably decreased in the NTx group compared to the control group and notably increased in the PD-MSCs cocultivation group. Their expression was significantly reduced in PD-MSCs cocultivation with the BIO group compared to the PD-MSCs cocultivation group (Figure 4A; * *p* < 0.05). Also, their expression was similar in the NTx group and IWP2 group. In contrast, the expression of these mRNA was remarkably increased in the PD-MSCs cocultivation with IWP2 treatment compared to the IWP2 group. PD-MSCs restored the mRNA expression of *IL-10* suppressed by IWP2 treatment in ovarian tissues (Figure 4C; * *p* < 0.05). The mRNA expression of *IL-6* showed a significant increase in the NTx group compared to the control group and remarkably decreased in the PD-MSCs cocultivation group compared to the NTx group. Their expression was notably increased in PD-MSCs cocultivation with the BIO group compared to the PD-MSCs cocultivation group (Figure 4B; * *p* < 0.05). Also, their expression was increased in the IWP2 group compared to PD-MSCs cocultivation (Figure 4D; * *p* < 0.05). These results suggest that inflammation induced by inhibiting *GSK3β* and *LRP6* in ovarian tissues is ameliorated by PD-MSCs.

We assessed the expression of *Wnt* signaling-related markers in ovarian tissues that were cocultivated with PD-MSCs and treated with BIO, which is *GSK3β* inhibitor, to determine the effect of PD-MSCs on *Wnt* signaling. The mRNA expression of *GSK3β* was decreased in the NTx group compared to the control group. The mRNA expression of *GSK3β* was markedly increased in the PD-MSCs cocultivation group compared to the NTx group. Also, their expression was considerably decreased in the PD-MSCs cocultivation with the BIO treatment group compared to the PD-MSCs cocultivation group (Figure 4E; * *p* < 0.05). Furthermore, the mRNA expression of *β-catenin* was considerably decreased in the NTx group compared to the control group and notably increased in the PD-MSCs cocultivation group compared to the NTx group. In addition, their expression was considerably decreased in the PD-MSCs cocultivation with the BIO treatment group compared to the PD-MSCs cocultivation group (Figure 4F; * *p* < 0.05).

To determine the effect of PD-MSCs on *Wnt* signaling, we examined the expression of factors related to the *Wnt* signaling pathway in ovarian tissues that were cocultivated with PD-MSCs and treated with IWP2, which is an *LRP* inhibitor, as a *Wnt* inhibitor. The mRNA expression of *LRP6* was notably decreased in the NTx group compared to the control group and considerably increased in the PD-MSCs cocultivation group. Their expression in the IWP2 group was similar to the NTx group. However, in the PD-MSCs cocultivation with the IWP2 treatment group, the mRNA expression of *LRP6* was recovered compared to the IWP2 group (Figure 4G; * *p* < 0.05). In addition, the mRNA expression of *β-catenin* was significantly decreased in the NTx group compared to the control group and significantly increased in the PD-MSCs cocultivation group compared to the NTx group. The mRNA expression of *β-catenin* was similar in the NTx group and the IWP2 treatment group. However, their expression recovered in the PD-MSCs cocultivation with the IWP2 treatment group compared to the IWP2 group (Figure 4H; * *p* < 0.05). These results imply that PD-MSCs activate *Wnt* signaling suppressed by inhibitors.

To evaluate whether *Wnt* signaling affects follicular development, we analyzed a factor related to follicular maturation (e.g., *BMP15*, *EGFR*). The mRNA expression of *BMP15* was significantly decreased in the NTx group compared to the control group and considerably increased in the PD-MSC cocultivation group compared to the NTx group. Their expression was remarkably decreased in the PD-MSC cocultivation with the BIO treatment group compared to the PD-MSC cocultivation group (Figure 4I; * *p* < 0.05). Also, IWP2 was treated to ovarian tissues, their expression was similar to the NTx group. However, the mRNA expression of *BMP15* was significantly increased in the PD-MSCs cocultivation with IWP2 treatment compared to the IWP2 group (Figure 4K; * *p* < 0.05). The mRNA expression of *EGFR* was significantly decreased in the NTx group compared to the control group and notably increased in the PD-MSC cocultivation group compared to the NTx group. Their expression was significantly decreased in the PD-MSC cocultivation with the BIO treatment group compared to the PD-MSC cocultivation group (Figure 4J; * *p* < 0.05). In addition, when IWP2 was treated to ovarian tissues, the mRNA expression of *EGFR* was similar to the NTx group. However, their expression was notably increased in the PD-MSCs cocultivation with IWP2 treatment compared to the IWP2 group (Figure 4L; * *p* < 0.05). PD-MSCs restore the expression of factors (e.g., *BMP15*, *EGFR*) related to follicular maturation inhibited by BIO and IWP2 treatment. These results suggest that *Wnt* signaling regulates follicular maturation in ovarian tissues.

### 3.5. Effect of PD-MSCs on Vascular Remodeling in LPS-Treated HUVECs (In Vitro)

Additionally, we analyzed vascular function in LPS-treated HUVECs to confirm that PD-MSCs improve vascular function. To demonstrate the impact of PD-MSCs on vascular function, we performed in vitro experiments using HUVECs treated with LPS to induce inflammation. PD-MSCs were cocultured with the HUVECs using an insert system for 24 h (Figure 5A). The tube-formation capacity of the endothelial cells was observed, which revealed few bridges and branch points in the LPS-treated group without cocultivation with PD-MSCs. After coculture with PD-MSCs, the number of bridges was enhanced compared to that in the LPS-treated group (Figure 5C,D; * *p* < 0.05). Moreover, we analyzed vascular permeability through a dextran assay to determine that PD-MSCs decrease vascular permeability (Figure 5B). Vascular permeability was significantly increased in the LPS-treated group without cocultivation with PD-MSCs compared to the control group and was significantly decreased in the LPS-treated group with cocultivation with PD-MSCs (Figure 5E; * *p* < 0.05). These data indicated that PD-MSCs enhance vascular function through regulation of endothelial cell permeability.

## 4. Discussion

The vascular network functions in the perfusion of nutrients, materials and blood, so it is an essential component in the metabolism of all tissues in the body and restoration of tissue function [40]. The formation of vessels can be divided into five stages: (i) tissue signaling, (ii) pericyte detachment, (iii) endothelial sprouting, (iv) pericyte recruitment, and (v) vascular remodeling and maturation. Recently, angiogenesis has been considered therapeutically as a crucial factor that improves the function of blood vessels in ischemic tissue through the delivery of proangiogenic factors [41]. On the other hand, excessive angiogenesis is detrimental to organs, causing inflammation, fibrosis and cancer, and blood vessels can provide nutrients required for the survival and proliferation of nearby malignant cells [42]. Recently, many researchers have focused on both positive and negative regulatory factors involved in therapeutic angiogenesis that closely regulate vascular function [43]. Therefore, various methods have been used to analyze the structure of vessels and blood flow function to evaluate vascular function [44]. For this reason, we hypothesized that improved vascular function through vascular remodeling rather than angiogenesis in injured ovarian tissues would improve ovarian function. Increased vascular permeability, which occurs via mechanisms including vascular barrier disintegration and leakage, is a progressive symptom that is mainly seen in chronic inflammation and various diseases [45]. Gerdes et al. demonstrated that ovarian vascular permeability increases as the number of leukocytes increases during ovulation [46]. This means that vascular permeability increases when inflammation occurs in ovarian tissues.

Inflammatory reactions in blood vessels cause damage to the endothelial cells of the vessels. As proinflammatory factors are transported through the bloodstream to various organs, they damage the function and structure of tissues [47,48]. The inflammatory response is a process involved in vascular remodeling, although inflammasome activation and pyroptosis cause endothelial cell senescence [49,50]. Additionally, the immune system is an essential mediator of ovarian function. According to Nakao et al., *TNF-α* inhibits ovulation by stimulating FSH to stimulate aromatase activity and progestin secretion [51]. Galvao et al. demonstrated that IL-1, which is a proinflammatory factor, inhibits progestin secretion and decreases the expression of luteinizing hormone (LH) receptors in granulosa cells. In particular, the authors reported that ovarian steroidogenesis is regulated by proangiogenic factors [52]. However, the mechanisms by which ovarian vasculature relates to the immune and inflammatory systems of the ovary remain uncertain.

MSCs have recently been used as a treatment for degenerative diseases to overcome the limitations of existing chemical treatments [53,54]. MSCs were reported to secrete proangiogenic factors, which regulate follicular development, through paracrine activity. Ovarian function is improved by the paracrine effect of growth factors such as *HGF* and *VEGF* secreted by MSCs [55,56]. Wulff et al. demonstrated that follicular development was decreased by the inhibition of *VEGF*, a proangiogenic factor [57]. In particular, in our previous report, we demonstrated that transplanted PD-MSCs improved ovarian function through enhanced vascular function via activation of the *PDGF* signaling pathway in rats with ovarian injury [33]. These therapeutic effects are maximized by optimization of the cell number and method of PD-MSC transplantation [58]. In this study, we evaluated that *HGF* secreted by PD-MSCs can improve ovarian function by restoring vascular permeability in damaged ovarian tissues.

In ovarian tissues, the regulation of *Wnt* signaling by MSCs also improves ovarian function. According to El-Derany et al., bone marrow-derived mesenchymal stem cells (BM-MSCs) exerted potential therapeutic effects via the *Wnt*/*β-catenin* pathway in the ovarian tissues of rats with radiotherapy-induced POF. The *Wnt*/*β-catenin* pathway improved follicular development and maturation by regulating the apoptosis and proliferation of follicles in the ovarian tissues of POF rat model [59]. However, the mechanism by which *Wnt* signaling activation by MSCs improves follicular development has not been clearly reported. In our study, we demonstrated that increased *HGF* induced by PD-MSCs triggers *Wnt* signaling in the vasculature of injured ovarian tissues of an ovariectomized rat model and induces follicular development by decreasing vascular permeability and inflammation in the ovary. This is the first finding that *HGF*-activated *Wnt* signaling improves ovarian function by decreasing vascular permeability.

## 5. Conclusions

PD-MSCs activate *Wnt* signaling by secretion of *HGF* in ovarian tissues of ovariectomized rat model. After transplanted PD-MSCs, the vascular structure in injured ovarian tissues was similarly changed to the normal group. The vascular permeability among *HGF* and *Wnt* signaling were decreased in Tx groups compared to the NTx group. Also, we confirm that cocultivation PD-MSCs with LPS-treated HUVECs decreased vascular permeability and enhanced vascular tube formation. Furthermore, secretion of AMH and follicular development were improved in the Tx group compared to the NTx group. Follicular maturation was improved in LPS treated-KGN with cocultivation PD-MSCs groups compared LPS treated-KGN groups.

Finally, to confirm the effect of *Wnt* signaling by PD-MSCs in ovarian tissues, we analyzed the expression of *Wnt* signaling-related genes and follicular maturation-related genes in ovarian tissues with BIO-treated PD-MSCs cocultivation or IWP2-treated PD-MSCs cocultivation. The cocultivation PD-MSCs with explanted ovarian tissues improved follicular maturation compared to cocultivation *Wnt* inhibitor treated-PD-MSCs with explanted ovarian tissues.

In conclusion, our findings indicate that increased *HGF* induced by PD-MSCs improved ovarian function via dual effects, namely by regulating vascular permeability and inhibiting inflammation, via activated *Wnt* signaling in rats with ovarian failure. Therefore, these findings provide new perspectives for the application of stem cell therapy in reproductive medicine.

## Figures and Tables

**Figure 1 cells-12-02708-f001:**
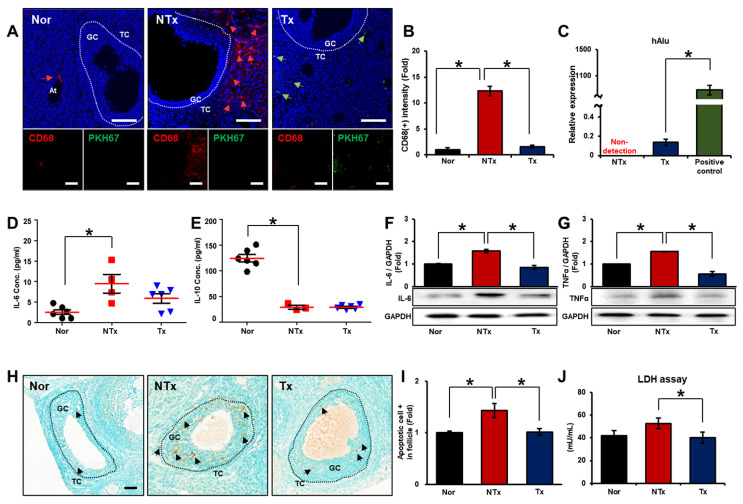
Anti-inflammatory effect of transplanted PD-MSCs in the ovarian tissues of half ovariectomized rats. (**A**) The gene expression of *CD68* and the localization of PKH67-labeled PD-MSCs were analyzed by immunofluorescence staining in ovaries after transplantation. (**B**,**C**) The intensity of positive *CD68* and positive PKH67 staining was analyzed with the program ImageJ. (**D**) The mRNA expression of *IL-6* and (**E**) *IL-10* was analyzed by qRT-PCR. (**F**) The gene expression of *IL-6* and (**G**) *TNFα* was analyzed by western blotting. (**H**) Apoptosis in the ovaries after transplantation was detected by measuring DNA fragmentation with a TUNEL assay. (**I**) The intensity of apoptosis detected in the antral follicles was quantified with the program 3D HISTECT. (**J**) The level of LDH was analyzed by ELISA. Abbreviations: NTx: nontransplantation, Tx: transplantation, GC: granulosa cell, TC: theca cell, At: atresia follicle, V: vessel, *IL-6*: interleukin-6, *IL-10*: interleukin-10, LDH: lactate dehydrogenase. The data represent three independent experiments and are presented as the mean ± S.D. Statistical significance was assessed using one-way ANOVA, with indicating * *p* < 0.05.

**Figure 2 cells-12-02708-f002:**
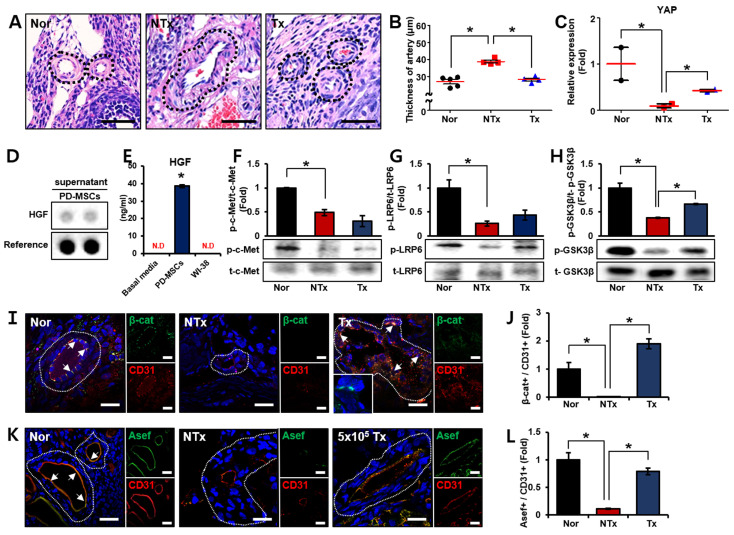
Increased *HGF* secreted by transplanted PD-MSCs improved vascular remodeling via *Wnt* signaling in the ovarian tissues of half-ovariectomized rats. (**A**) The structure of the ovarian arteries was analyzed with the program 3D HISTECT. (**B**) The thickness of the artery was quantified with the program 3D HISTECT. (**C**) The mRNA expression of *YAP* was analyzed by qRT-PCR. (**D**) *HGF* dot blot analysis was carried out by RTK assay, and (**E**) the *HGF* concentration was analyzed by ELISA. (**F**) The gene expression levels of *c-Met*, (**G**) *LRP6* and (**H**) *GSK3β* were analyzed by western blotting. (**I**) The localization of *β-catenin* and *CD31* in the ovarian arteries were analyzed by immunofluorescence. (**J**) The intensity of positive *β-catenin* staining was analyzed with the program ImageJ. (**K**) The localization of *Asef* and *CD31* in the ovarian arteries were analyzed by immunofluorescence. (**L**) The intensity of positive *Asef* staining was analyzed with ImageJ program. Abbreviations: NTx: nontransplantation, Tx: transplantation, PD-MSCs: placenta-derived mesenchymal stem cells, *YAP*: yes1 associated transcriptional regulator, *HGF*: hepatocyte growth factor, *LRP6*: LDL receptor-related protein 6, *GSK3β*: glycogen synthase kinase 3 beta, *Asef*: Rho guanine nucleotide exchange factor. The data represent three independent experiments and presented as the mean ± S.D. Statistical significance was assessed using one-way ANOVA, with indicating * *p* < 0.05.

**Figure 3 cells-12-02708-f003:**
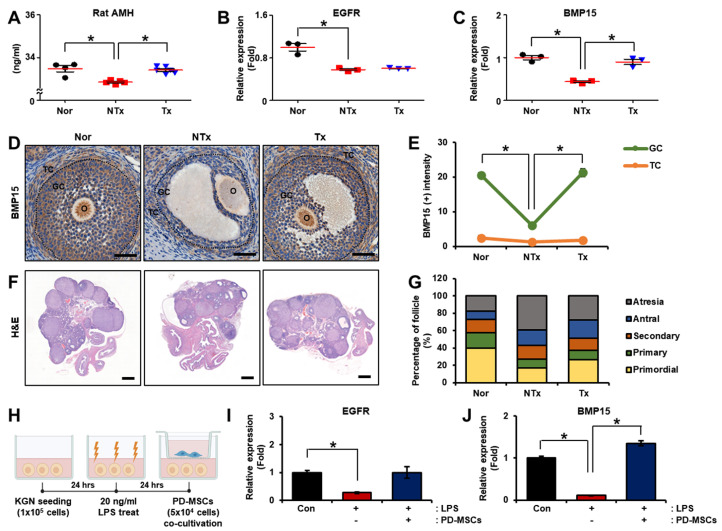
Effect of PD-MSCs on follicular development in the ovarian tissues of half ovariectomized rats and LPS-treated KGN cells. (**A**) The level of AMH in individual serum samples was examined by ELISA. (**B**) The mRNA expression of *EGFR* and (**C**) *BMP15* in the ovary was assessed using qRT-PCR. (**D**) The localization and gene expression of *BMP15* in ovarian tissues were analyzed by immunohistochemistry staining. (**E**) The intensity of *BMP15* staining in ovarian follicles was analyzed with the program 3D HISTECH. Scale bar: 100 µm, magnification: 20×. (**F**) Histological examination of follicular development in the ovary was conducted by H&E staining. (**G**) The percentages of follicles at different stages of follicular development were analyzed with the program 3D HISTECH. Scale bar: 500 µm, magnification: 2×. (**H**) Schematic diagram showing the cocultivation of LPS-treated KGN cells with PD-MSCs. (**I**) The mRNA expression of *EGFR* and (**J**) *BMP15* was analyzed by qRT-PCR. Abbreviations: NTx: nontransplantation, Tx: transplantation, AMH: anti-mullerian hormone, *EGFR*: epidermal growth factor receptor, *BMP15*: bone morphogenetic protein 15, LPS: lipopolysaccharide. The data represent three independent experiments and are presented as the mean ± S.D. Statistical significance was assessed using one-way ANOVA, with indicating * *p* < 0.05.

**Figure 4 cells-12-02708-f004:**
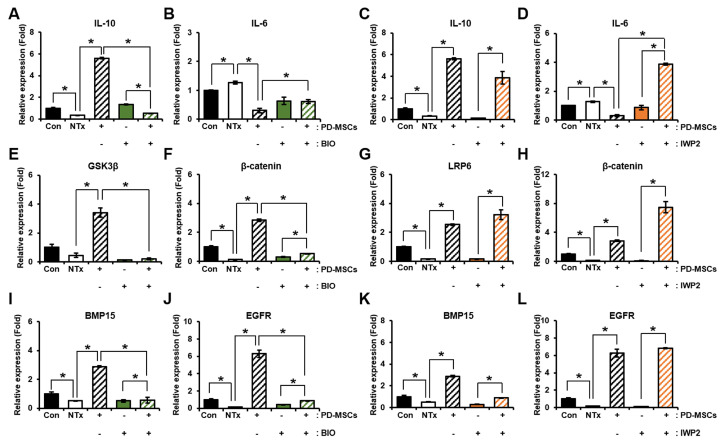
Effect of PD-MSCs on vascular remodeling and follicular development via *Wnt* signaling in Wnt inhibitor-treated ovaries (ex vivo). (**A**) The mRNA expression of *IL-10* and (**B**) *IL-6* in the cocultivation of explanted ovarian tissue system with PD-MSCs and BIO treatment was assessed using qRT-PCR. (**C**) The mRNA expression of *IL-10* and (**D**) *IL-6* in the cocultivation of explanted ovarian tissue system with PD-MSCs and IWP2 treatment was assessed using qRT-PCR. (**E**) The mRNA expression of *GSK3β* and (**F**) *β-catenin* in the cocultivation of explanted ovarian tissue system with PD-MSCs and BIO treatment was assessed using qRT-PCR. (**G**) The mRNA expression of *LRP6* and (**H**) *β-catenin* in the cocultivation of explanted ovarian tissues system with PD-MSCs and IWP2 treatment was assessed using qRT-PCR. (**I**) The mRNA expression of *BMP15* and (**J**) *EGFR* in the cocultivation of explanted ovarian tissues system with PD-MSCs and BIO treatment was analyzed by qRT-PCR. (**K**) The mRNA expression of *BMP15* and (**L**) *EGFR* in the cocultivation of explanted ovarian tissues system with PD-MSCs and IWP2 treatment was analyzed by qRT-PCR. Abbreviations: Con: control, NTx: non-cocultivation, PD-MSCs: placenta-derived mesenchymal stem cells, *IL-6*: interleukin-6, *IL-10*: interleukin-10, *GSK3β*: glycogen synthase kinase 3 beta, *LRP6*: low-density lipoprotein receptor-related protein 6, *EGFR*: epidermal growth factor receptor, *BMP15*: bone morphogenetic protein 15. The data represent three independent experiments and presented as the mean ± S.D. Statistical significance was assessed using one-way ANOVA, with indicating * *p* < 0.05.

**Figure 5 cells-12-02708-f005:**
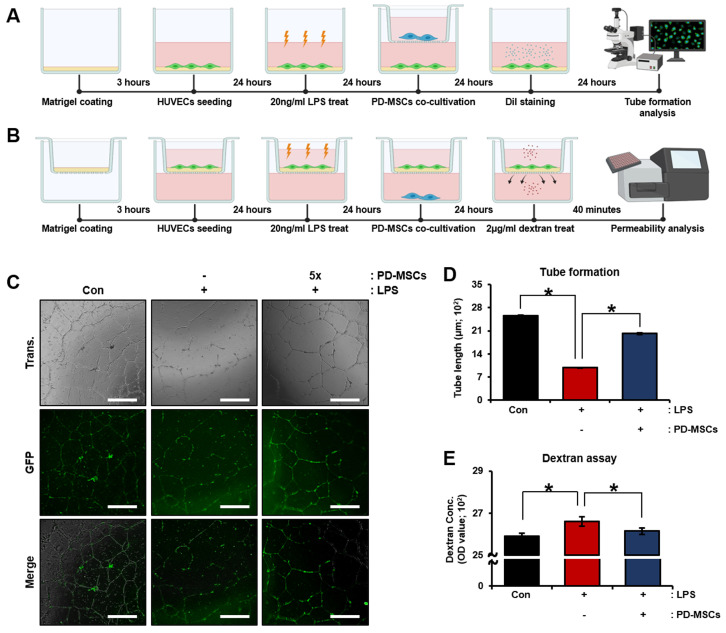
Effect of PD-MSCs on vascular function in LPS-treated HUVECs (in vitro). (**A**,**B**) Schematic diagram describing the cocultivation of LPS-treated HUVECs with PD-MSCs (**C**) The tube formation capacity of HUVECs was analyzed using Dil staining. (**D**) The vascular length of HUVECs was analyzed Image J program. (**E**) The vascular permeability of HUVECs was analyzed by dextran assay. Scale bar: 500 µm, magnification: 4X. Abbreviations: Con: control, NTx: non-cocultivation, PD-MSCs: placenta-derived mesenchymal stem cells, LPS: lipopolysaccharide. The data are representative of three independent experiments and expressed as the mean ± S.D. Statistical significance was determined by using one-way ANOVA, * *p* < 0.05.

**Table 1 cells-12-02708-t001:** Comparison of follicle counts after transplantation in vivo.

	Primordial	Primary	Secondary	Antral	Atresia
Normal (n = 4)	184 ± 32.07	40.25 ± 7.79	54.25 ± 10.90	51.25 ± 9.29	38.75 ± 6.10
NTx (n = 3)	101 ± 10.15 *	26.33 ± 7.00	51.00 ± 8.14	39.67 ± 5.17	97.67 ± 6.36 *
Tx (n = 3)	185.67 ± 12.17 **	47.67 ± 6.69 **	56.67 ± 5.61	61.33 ± 3.84 **	36.33 ± 3.71 **

*: NTx vs. Nor (*p* < 0.05); **: Tx vs. NTx (*p* < 0.05).

## Data Availability

Data are contained within the article.

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
