# Peer review of "Increased Hepatocyte Growth Factor Secretion by Placenta-Derived Mesenchymal Stem Cells Improves Ovarian Function in an Ovariectomized Rat Model via Vascular Remodeling by Wnt Signaling Activation"

_cells, 2023, doi:10.3390/cells12232708_

Round 1

Reviewer 1 Report (Previous Reviewer 1)

Comments and Suggestions for Authors

The protocol of experiments have not been adequately described in Abstract, whilst the analytical methods retained not be indicated at all. If these weak points would be corrected, I believe, the manuscript could be suitable for publication.

Comments on the Quality of English Language

I am not a specialist, but I feel, that some expressions are inadequate. I believe, these expressions could be corrected by the Publisher during preparation the Galley Proof.

Author Response

Point #1: The protocol of experiments have not been adequately described in Abstract, whilst the analytical methods retained not be indicated at all. If these weak points would be corrected, I believe, the manuscript could be suitable for publication.

  • Author’s response:

Thank you for your critical comments. As your comment, we modified the abstract to describe the experiment protocol in more detail.

Reviewer 2 Report (Previous Reviewer 2)

Comments and Suggestions for Authors

  The Reviewer is generally satisfied with the comments made by the authors. However, Figure 2 still lacks clarity. How can the expression of the genes be detected by immunofluorescence? My guess would be that there are  the proteins encoded by these genes. In Figure 2, the images presented in   (I) and  (K) look like the convenient immunofluorescent detection of certain antigens.   

Author Response

Cells-2707072

Increased hepatocyte growth factor secretion by placenta derived-mesenchymal stem cells improves ovarian function in an ovariectomized rat model via vascular remodeling by Wnt signaling activation

Author's Reply to the Review Report (Reviewer 2)

Point #1: The Reviewer is generally satisfied with the comments made by the authors. However, Figure 2 still lacks clarity. How can the expression of the genes be detected by immunofluorescence? My guess would be that there are  the proteins encoded by these genes. In Figure 2, the images presented in   (I) and  (K) look like the convenient immunofluorescent detection of certain antigens.    

  • Author’s response:

We appreciate the reviewer for pointing out an important point. Firstly, we apologize for any confusion. As you mentioned, our focus was on protein localization, not gene immunofluorescence analysis. In Figure 2, (I) depicts the co-staining of β-catenin and CD31, and (K) depicts the co-staining of Asef and CD31. Following this, we conducted an analysis of the localization of β-catenin and Asef in relation to CD31.

Therefore, we have revised the sentence in the manuscript to avoid confusion: '(I) The localization of β-catenin and CD31 in the ovarian arteries was analyzed by immunofluorescence staining.' and '(K) The localization of Asef and CD31 in the ovarian arteries was analyzed by immunofluorescence staining.'

Fig. 2

This manuscript is a resubmission of an earlier submission. The following is a list of the peer review reports and author responses from that submission.

Round 1

Reviewer 1 Report

Comments and Suggestions for Authors

The authors have done a large work and gained numerous results, which could be interesting from both btheoretical and medicinal viewpoints. Nevertheless, theis presentation requires substantial improvement to be easy to understand.

TITLE: the abbreviations should be explained, otherwise the title is not understandable for the majority of readers. The influence of HGF to some genes related to WNT signalling does not means, that „HGF improves ovarian function via vascular remodeling by Wnt signaling activation“. The authors might claim only influence of HGF on WNT signalling.

ABSTRACT. „. However, therapeutic mechanism between vascular remodeling and ovarian function still unclear.“ Please formulate the concrete hypothesis to be validated by the author’s studies. Why just  „therapeutic“ but not general biological mechanisms? The protocol of the experiments and the analytical methods used should be added. What for ovarian disfunctions was used as model? How it was mitigated by HGF? How the author’s hypothesis was supported? The conclusion like „These finding offer new insight ... and ...new avenues“ is not concrete and, therefore, not informative.

INTRODUCTION containes too many irrelevant information, which should be deleted, and Introduction should be focused more to the concrete present study. On the other hand, the concrete hypothesis which the authors try to validate was poorly formulated . „Nevertheless, the mechanisms un derlying the effects of Wnt signaling on vascular permeability in the ovary remain uncertain.“ cannot be considered as concrete hypothesis. The reasons of the performed study and its novelty are not clear for the reader. The design of the study „to determine whether the expression of HGF by PD-MSCs improves ovarian function in an ovariectomized rat model via vascular remodeling by Wnt signaling activation“ including short description of experiments and analytical methods described in Materials and Methods in details should be shortly outlined in Introduction. Otherwise the correctness of the author’s approach is difficult to assess. This description should be clearly linked with all the the Results.

MATERIALS AND METHODS. Please add the information concerning specificity, sensitivity, positive and negative control for each analytical method.

RERSULTS. Figure legends. Please explain the abbreviations. The reader should understand, what was done without reading the Main text.  TUNEL detects njclear DNA fragmentation, but not cytoplasmic/mitochondrial apoptosis. Therefore, the kind of apoptosis should be indicated throughout the manuscript.

DISCUSSION should be focused on the author’s hypothesis (pro- and contra-), on the novelty of the obtained data in comparizon with the previous reports. It should outlined the directions of the further studies in this area and their significance for praxis. The general and irrelevant information should be reduced.

CONCLUSIONS are shprt, too general and, therefore, not informative. Please formulate the concrete conclusions based on the concrete results described in the Results section.

GENERAL. English requires improvement.

Comments on the Quality of English Language

English requires improvement

Author Response

Cells-2488600

Increased HGF secretion by PD-MSCs improves ovarian function in an ovariectomized rat model via vascular remodeling by Wnt signaling activation

Hyeri Park, Dae Hyun Lee, Jun Hyeong You, Jin Seok, Ja-Yun Lim, Gi Jin Kim

Author's Reply to the Review Report (Reviewer 1)

Comments and Suggestions for Authors

The authors have done a large work and gained numerous results, which could be interesting from both btheoretical and medicinal viewpoints. Nevertheless, theis presentation requires substantial improvement to be easy to understand.

Author’s reply:

We greatly appreciate the reviewer’s positive statement. Also, we try to improve according to the reviewer’s comments.

Point #1: TITLE: the abbreviations should be explained, otherwise the title is not understandable for the majority of readers. The influence of HGF to some genes related to WNT signalling does not means, that „HGF improves ovarian function via vascular remodeling by Wnt signaling activation“. The authors might claim only influence of HGF on WNT signalling.

  • Author’s response:

Thank you for your critical comments. The abbreviation of the title was corrected to the full name as “Increased hepatocyte growth factor secretion by placenta derived-mesenchymal stem cells improve ovarian function in an ovariectomized rat model via vascular remodeling by Wnt signaling activation” as follows for readers to easily understand it.

Point #2: The influence of HGF to some genes related to WNT signalling does not means, that “HGF improves ovarian function via vascular remodeling by Wnt signaling activation”. The authors might claim only influence of HGF on WNT signalling.

  • Author’s response:

We greatly appreciate the reviewer bringing up this important point. It’s generally well known through many references that “HGF induces Wnt signaling” [Morishita, R et al., J Atheroscler Thromb 1998. Monga, S.P. et al., Cancer Res 2002.]. We confirmed that HGF was secreted by PD-MSCs and Wnt signaling was activated in ovarian tissues transplanted with PD-MSCs. In addition, we confirmed that the structure and permeability of vascular were improved in the ovarian tissues of the rat model transplanted with PD-MSCs, and finally, the follicular development was improved. To demonstrate that these therapeutic effect is due to activated Wnt signaling, we co-cultured PD-MSCs treated with Wnt inhibitor in explanted ovarian tissues. According to Figure. 4, PD-MSCs activate Wnt signaling and improve ovarian follicle maturation. However, the expression of genes related to follicle maturation decreased when Wnt signaling was inhibited. These results means that HGF improves ovarian function via vascular remodeling by Wnt signaling activation.

  • Morishita, R.; Nakamura, S.; Hayashi, S.; Aoki, M.; Matsushita, H.; Tomita, N.; Yamamoto, K.; Moriguchi, A.; Higaki, J.; Ogihara, T. Contribution of a vascular modulator, hepatocyte growth factor (HGF), to the pathogenesis of cardiovascular disease. J Atheroscler Thromb 1998, 4, 128-134.
  • Monga, S.P.; Mars, W.M.; Pediaditakis, P.; Bell, A.; Mule, K.; Bowen, W.C.; Wang, X.; Zarnegar, R.; Michalopoulos, G.K. Hepatocyte growth factor induces Wnt-independent nuclear translocation of beta-catenin after Met-beta-catenin dissociation in hepatocytes. Cancer Res 2002, 62, 2064-2071.

Point #3: ABSTRACT. „. However, therapeutic mechanism between vascular remodeling and ovarian function still unclear.“ Please formulate the concrete hypothesis to be validated by the author’s studies. Why just  „therapeutic“ but not general biological mechanisms? The protocol of the experiments and the analytical methods used should be added. What for ovarian disfunctions was used as model? How it was mitigated by HGF? How the author’s hypothesis was supported? The conclusion like „These finding offer new insight ... and ...new avenues“ is not concrete and, therefore, not informative.

  • Author’s response:

We appreciate the reviewer for pointing out an important point. As described in the paper, our hypothesis is “HGF by PD-MSCs improves ovarian function in an ovariectomized rat model via vascular remodeling by Wnt signaling activation.”. So, we constructed a rat model of ovarian dysfunction by cutting half of each ovary. To support the objective, we first analyzed HGF secretion by PD-MSCs. It was confirmed that Wnt signaling was activated in the ovary of rat model transplanted with PD-MSCs, and vascular remodeling and ovarian function were improved. Finally, ex vivo experiments using Wnt inhibitor was analyzed to confirm the efficacy of treatment by activation of Wnt signaling. In conclusion, we confirmed “HGF by PD-MSCs improves ovarian function in an ovariectomized rat model via vascular remodeling by Wnt signaling activation.”. We suggest that the therapeutic efficacy of PD-MSCs in a preclinical model can suggest potential as a stem cell therapy in a clinical setting.

Based on your comment, we added the experiments and revised the concrete conclusion.

Point #4: INTRODUCTION containes too many irrelevant information, which should be deleted, and Introduction should be focused more to the concrete present study. On the other hand, the concrete hypothesis which the authors try to validate was poorly formulated . „Nevertheless, the mechanisms un derlying the effects of Wnt signaling on vascular permeability in the ovary remain uncertain.“ cannot be considered as concrete hypothesis. The reasons of the performed study and its novelty are not clear for the reader. The design of the study „to determine whether the expression of HGF by PD-MSCs improves ovarian function in an ovariectomized rat model via vascular remodeling by Wnt signaling activation“ including short description of experiments and analytical methods described in Materials and Methods in details should be shortly outlined in Introduction. Otherwise the correctness of the author’s approach is difficult to assess. This description should be clearly linked with all the the Results.

  • Author’s response:

Thank you for your critical comments.

As you mentioned, we focused on our present study for the effect of HGF secreted by PD-MSCs on ovarian function in an ovariectomized rat model via vascular remodeling by Wnt signaling activation. (ß- containes too many irrelevant information, which should be deleted, and Introduction should be focused more to the concrete present study ) Also, the introduction has been amended to specifically mention the materials and methods to reduce reader confusion.

Point #5: MATERIALS AND METHODS. Please add the information concerning specificity, sensitivity, positive and negative control for each analytical method.

  • Author’s response:

Thank you for your critical comments. According reviewer’s comments, we revised more detailed materials and methods.

Point #6: RERSULTS. Figure legends. Please explain the abbreviations. The reader should understand, what was done without reading the Main text.  TUNEL detects njclear DNA fragmentation, but not cytoplasmic/mitochondrial apoptosis. Therefore, the kind of apoptosis should be indicated throughout the manuscript.

  • Author’s response:

Thank you for your important comments. Following reviewer’s comments, we have added abbreviations to the figure legend. In addition, the text was modified to mention the kind of apoptosis in the results of the TUNEL assay.

Point #7: DISCUSSION should be focused on the author’s hypothesis (pro- and contra-), on the novelty of the obtained data in comparizon with the previous reports. It should outlined the directions of the further studies in this area and their significance for praxis. The general and irrelevant information should be reduced.

  • Author’s response:

Thank you for your critical comments. According to reviewer’s comments, we modified it to reduce irrelevant information and focus on the novelty of the data obtained by comparison with previous reports.

  • 14, L576-582, “In particular, in our previous report, ~~~ HGF secreted by PD-MSCs can improve ovarian function by restoring vascular permeability in damaged ovarian tissues”
  • 14-15, L590~594, “In our study, we demonstrated that~~~~ HGF-activated Wnt signaling improves ovarian function by decreasing vascular permeability”

Point #8: CONCLUSIONS are short, too general and, therefore, not informative. Please formulate the concrete conclusions based on the concrete results described in the Results section.

  • Author’s response:

Thank you for your critical comments. We modified the CONCLUSIONS to describe it in more detail.

Point #9: GENERAL. English requires improvement.

  • Author’s response:

Thank you for your critical comments. We checked English proofreading once more.

Reviewer 2 Report

Comments and Suggestions for Authors

At a moment, MSCs  from different sources  are considered as an attractive tool for cell therapy.  In this MS, the  authoer have demonstrated that the intravenous application of human PD-MSCs   improved ovarian  function in an ovariectomized rat model via vascular remodeling by Wnt signaling activation. They supposed that the elevation of HGF due to these MSCs could contribute to this improvement.

The study is well done, a lot of contemporary experimental approaches are involved to support the authors’ considerations.

Meanwhile, some points need to be addressed.

General Comments

Why did the authors use human PD-MSCs instead of rat? It seems more adequate to use allogeneic, instead of   xenogeneic tissue source.

The MSC secretome includes a lot of anti-inflammatory and angiogenic mediators. The authors have cited their own paper on VEGF effects. Why did the authors are focused on HGF? Do MSCs secrete another  molecules involved in Wnt-path regulation?

It seems to Reviewer, that demonstrate the particular role of MSC-derived HGF it would be to correct to inhibit HGF secretion by MSCs and use HGF-deficient MSCs.

Why did the authors imply systemic administration of MSCs instead of local? How long the effect of MSC injection was occur?

Currently, soluble secreted components of MSCs are positioned as a valuable cell therapy tool, comparable in the efficacy to MSCs themselves, but safer. Please, argue your choice of whole cells application.

Specific comments

L 284-285 «Additionally, we analyzed the expression of genes related to inflammation in the serum …»  How is it possible to measure gene expression in the serum? Please, clarify.

L350-351” The localization and gene expression of β-catenin and CD31 in the ovarian arteries were analyzed by immunofluorescence staining…” Wgere are the data on gene expression of β-catenin and CD31 and how it is possible to do with immunofluorescence staining?

Author Response

Cells-2488600

Increased HGF secretion by PD-MSCs improves ovarian function in an ovariectomized rat model via vascular remodeling by Wnt signaling activation

Hyeri Park, Dae Hyun Lee, Jun Hyeong You, Jin Seok, Ja-Yun Lim, Gi Jin Kim

Author's Reply to the Review Report (Reviewer 2)

Comments and Suggestions for Authors

At a moment, MSCs  from different sources  are considered as an attractive tool for cell therapy.  In this MS, the  authoer have demonstrated that the intravenous application of human PD-MSCs   improved ovarian  function in an ovariectomized rat model via vascular remodeling by Wnt signaling activation. They supposed that the elevation of HGF due to these MSCs could contribute to this improvement.

The study is well done, a lot of contemporary experimental approaches are involved to support the authors’ considerations.

Meanwhile, some points need to be addressed.

Author’s reply:

We greatly appreciate the reviewer’s positive statement that “The study is well done, a lot of contemporary experimental approaches are involved to support the authors’ considerations.”.

General comments

Point #1: Why did the authors use human PD-MSCs instead of rat? It seems more adequate to use allogeneic, instead of  xenogeneic tissue source.

  • Author’s response:

We appreciate the reviewer for pointing out an important point. Our project is to develop the stem cell therapeutics for patients with several degenerative diseases. So, their therapeutic efficacies were validated on animal diseases before human clinical trials. Especially, human PD-MSCs have strongly immunomodulatory effect [Chang et al., Stem Cells 2006.]. Also, according to our previous report, the engraftment of PD-MSCs was analyzed about 1 week after transplantation and then disappeared [Seok et al., Antioxidants 2020]. The advantages and homing of PD-MSCs suggest that they can prove the safety of xenograft. As well as, recently, many scientists reported that “Xenogeneic stem cell transplantation has therapeutic effects and broad application prospects in various degenerative diseases.” [Jiang et al., World J Clin Cases 2021.]. Therefore, we demonstrated that transplantation of human PD-MSCs into rats has therapeutic efficacy in a safe system.

  • Chang, C.; Yen, M.; Chen, Y.; Chien, C.; Huang, H.; Bai, C.; Yen, B. L. Placenta-derived multipotent cells exhibit immunosuppressive properties that are enhanced in the presence of interferon-gamma. Stem Cells 2006, 24(11), 2466-77.
  • Seok, J.; Park, H.; Choi, J.H.; Lim, J.; Kim, K.G.; Kim, G.J. Placenta-derived mesenchymal stem cells restore the ovary function in an ovariectomized rat model via an antioxidant effect. Antioxidants 2020, 9(7), 591.
  • Jiang, L.; Li, H.; Liu, L. Xenogeneic stem cell transplantation: Research progress and clinical prospects. World J Clin Cases 2021, 9(16), 3826-3837.

Point #2: The MSC secretome includes a lot of anti-inflammatory and angiogenic mediators. The authors have cited their own paper on VEGF effects. Why did the authors are focused on HGF? Do MSCs secrete another  molecules involved in Wnt-path regulation?

  • Author’s response:

We greatly appreciate the reviewer bringing up this important point. As the reviewer commented, MSCs are well known to secrete various cytokines involved in anti-inflammation and angiogenesis.

In previous reports, we analyzed cytokine arrays to identify cytokines secreted by PD-MSCs (Figure. 1). From the cytokine array results, it was confirmed that HGF has a strong expression compared to other cytokines related to angiogenesis and anti-inflammation. In addition, it is known that HGF-induced Wnt signaling regulates anti-inflammatory and vascular remodeling.

Figure.1 Cytokine array of PD-MSCs

Point #3: It seems to Reviewer, that demonstrate the particular role of MSC-derived HGF it would be to correct to inhibit HGF secretion by MSCs and use HGF-deficient MSCs.

  • Author’s response:

We greatly appreciate the reviewer bringing up this important point. We only performed ex vivo experiments on inhibition of Wnt signaling. We agree with the reviewer's comment that experiments to inhibit HGF secretion by MSCs are also necessary. However, since we have a limitation that the period for conducting additional experiments is too short (5 days), the proof of the inhibition of HGF secretion by MSCs will be studied in depth in the next paper.

Point #4: Why did the authors imply systemic administration of MSCs instead of local? How long the effect of MSC injection was occur?

  • Author’s response:

Thank you for your critical comments. In our previous report, we demonstrated that PD-MSCs were transplanted into the ovary, ovarian function was improved by antioxidant effect [Park et al., Int J Mol Sci 2022.]. When PD-MSCs were transplanted into the ovary directly, transplanted PD-MSCs were significantly increased engraftment compared to i.v. injection by tail vein and improved their therapeutic effect including antioxidant effect. However, in the case of the rat model analyzed in this paper, since both ovaries were half-ovariectomized, intravenous transplantation was performed because intraovarian transplantation could damage the ovaries.

  • Park, H.; Seok, J.; You, J.H.; Lee, D.H.; Lim, J.Y.; Kim, G.J. Can a Large Number of Transplanted Mesenchymal Stem Cells Have an Optimal Therapeutic Effect on Improving Ovarian Function? Int J Mol Sci 2022, 23.

Point #5: Currently, soluble secreted components of MSCs are positioned as a valuable cell therapy tool, comparable in the efficacy to MSCs themselves, but safer. Please, argue your choice of whole cells application.

  • Author’s response:

Thank you for your important comments. As reviewer’s comment, exosome of MSCs have potential in stem cell therapy research because it has the advantage of immune-compatibility and non-cytotoxic. However, exosome of MSCs still have the limitation of isolation technologies, purification and quality control. However, the advantages of MSCs including PD-MSCs therapy have several therapeutic potentials (e.g., differentiation and cell-cell contact signaling).

Specific comments

Point #6: L284-285 «Additionally, we analyzed the expression of genes related to inflammation in the serum …»  How is it possible to measure gene expression in the serum? Please, clarify.

  • Author’s response:

Thank you for your critical comments. We analyzed the concentration of inflammatory markers related to inflammation in the serum and corrected it in the revised manuscript.

Point #7: L350-351” The localization and gene expression of β-catenin and CD31 in the ovarian arteries were analyzed by immunofluorescence staining…” Wgere are the data on gene expression of β-catenin and CD31 and how it is possible to do with immunofluorescence staining?

  • Author’s response:

Thank you for your critical comments. In Figure.2I, we were able to analyze the localization of β-catenin and CD31, and the intensity of each gene by immunofluorescence staining in ovarian arteries. We analyzed the intensity of each gene through the Image J program, and the ratio of β-catenin to CD31 (Figure 2J).
